# Subjective Perception and Psychoacoustic Aspects of the Laryngectomee Voice: The Impact on Quality of Life

**DOI:** 10.3390/jpm13030570

**Published:** 2023-03-22

**Authors:** Massimo Mesolella, Salvatore Allosso, Roberto D’aniello, Emanuela Pappalardo, Vincenzo Catalano, Giuseppe Quaremba, Gaetano Motta, Grazia Salerno

**Affiliations:** 1Unit of Otorhinolaryngology, Department of Neuroscience, Reproductive Sciences and Dentistry, Federico II University of Naples, 80131 Naples, Italy; 2Department of Advanced Biomedical Sciences, Federico II University of Naples, 80131 Naples, Italy; 3Unit of Otorhinolaryngology, University Luigi Vanvitelli, 80138 Naples, Italy

**Keywords:** total laryngectomy, substitution voice, INFVo scale, quality of life

## Abstract

Purpose: A retrospective study is presented to correlate the inter-judge consistency for the different psycho-perceptual parameters of the recently proposed Impression Noise Fluency Voicing (INFVo) perceptual rating scale for substitution voices, and the vocal function as perceived by the patient. Methods: The scale Voice-Related Quality of Life (V-RQoL) and the Self Evaluation of Communication Experiences After Laryngectomy scale (SECEL)—a self-evaluation questionnaire of communicative experience after laryngectomy surgery—were administered to 89 total laryngectomees, subdivided in four groups depending on their type of alaryngeal voice (i.e., tracheoesophageal and esophageal speakers, electro larynx users, voiceless patients), in order to evaluate the impact of the impairment of the phonatory function on the quality of life. Results: No significant differences exist among the various groups on their perception of QoL using subjective questionnaires, whereas the INFVo scale has proven to be a useful tool for the description and analysis of the psychoacoustic characteristics of the vocal signal and a reliable instrument to correctly classify the patients. It is also notable that the judgement of the patients on their own voice and those of the referees are highly significant. Conclusion: Although speech rehabilitation for the acquisition of a substitution voice offers a new way of communication for the laryngectomized patients, nonetheless, their QoL is not significantly related to the type of substitution voice. Therefore, improving the patient’s adaptation to the new phonatory condition is mandatory.

## 1. Introduction

The treatment of advanced laryngeal cancer has a severe impact on the quality of life of patients [1]. When total laryngectomy (TL) is the choice surgical procedure, the removal of the entire laryngeal organ seriously compromises many important physiological functions, such as breathing, speaking, swallowing, taste and smell. In addition, the subsequent profound changes to the face and the neck due to the creation of a permanent tracheostoma negatively affect the image of the patient himself and his self-confidence, remarkably impairing the quality of life (QoL) of the laryngectomee [2]. This is equally true for both physical and functional health, as it is for emotional and psychological feelings [3]. These are the main reasons why this surgical procedure is considered a devastating experience.

The loss of the voice, the most important means of personal interaction with the outside world, strongly impairs the communicative effectiveness of the laryngectomee, influencing his psychological state, level of independence and the maintenance of a satisfactory social life [4]. In fact, it is well known that laryngectomees tend to experience avoidance behaviors during social interactions, reducing previous relationships or even deciding to interrupt any further external contacts. Therefore, it is mandatory to restore speech as soon as possible, even during primary surgery—the insertion of a tracheoesophageal prosthesis (TEP), a rehabilitative technique that actually represents the most commonplace method of vocal rehabilitation in the world [5]. In tracheoesophageal speech, TEP prevents saliva or food from entering the lower respiratory tract and enables air to enter the esophagus, allowing the pharyngo-esophageal (PE) segment to vibrate.

Esophageal speech (ES) and electrolarynx speech (ELS) are two alternative methods of vocal rehabilitation suggested when physical or psychological contraindications to TEP rehabilitation exist [6,7]. In esophageal speech, the air is voluntarily introduced into the esophagus through the mouth, using different techniques, then subsequently expelled by causing mucosal vibrations in the neoglottis. The electrolarynx is an external sound source device that the patient can place against the neck or cheek [8]. Unfortunately, many patients, especially those living in poor socioeconomic conditions, often remain voiceless (no voice, NV). The role of the speech therapist, together with the other skillful members of the team that deal with the patients undergoing TL, is to assist and guide them on the choice of the most appropriate rehabilitative method for achieving a substitution voice [9]. To evaluate the substitution voice in laryngectomees, a multidimensional assessment based on acoustic analysis, perceptual evaluation and patient-reported outcomes (PROs) is recommended, and the administration of scales of evaluation of vocal psycho-perceptual parameters and self-perceptive scales are an important part of the study. In 2006, Moerman et al. [10,11] proposed a new scale for the perceptual evaluation and the rating of substitution voices, namely, the INFVo scale, which includes both voice and speech parameters and tries to score the following five parameters: impression, intelligibility, noise, fluency and voicing. Parameter I reflects the overall voice quality as well the impression of intelligibility; N deals with the amount of annoyance caused by the audibility of uncontrolled noises, such as bubbly and breathy noises produced during speech; F reflects the perceived smoothness of the sound production; and Vo indicates whether the voicing is supposed to be voiced or unvoiced. Each parameter is scored on a visual analogue scale from 0 (minimally deviant) to 10 (maximally deviant substitution voicing). This validated auditory-perceptual scale, the reliability of which has also been proven in native Italian-speaking patients [12,13,14,15,16], seems to overwhelm the limits of the previously proposed GRBAS perceptual rating scale (grade, roughness, breathiness, asthenia, strain, on a scale of 0–3, where 0 = normal, 1 = mild degree, 2 = moderate degree and 3 = high degree) [17]. Even if the GRBAS scale presented a positive correlation with acoustic, aerodynamic and voice-related quality of life questionnaires [18,19,20], it does not seem to be the best tool for evaluating substitution voices. In fact, since substitution speech differs from laryngeal speech in the nature of the source oscillator, it is often scores as G3, which means severely impaired [17,18,20,21,22]. Moreover, some perceptual features are unique for substitution voices, and consequently not included in the GRABS scale. The aim of the present work was to analyze the relationship between the various psycho-perceptual parameters evaluated using the INFVo scale and assessed by a multidisciplinary team of substitution voice experts, and the vocal function as perceived by the patient. This in order to stress the impact of the eventual impairment of the phonatory function on the quality of life. Objective and subjective vocal evaluations were compared, to clarify whether a relationship exists between the effectiveness of the speech therapy rehabilitation treatment and the degree of the patient satisfaction, mainly concerning improvement in the quality of daily life.

## 2. Materials and Methods

### 2.1. Clinical Series

Our clinical series consists of 89 total laryngectomees, 77 males and 12 females, from 44 to 81 years (mean age 63.5 ys, SD = 9.07. Associate postoperative radiotherapy and chemotherapy were performed in 39 patients (43.8%), whereas 33 (37%) underwent radiotherapy only. Only 58 laryngectomees (65%) attended speech therapy sessions after surgery. Patient exclusion criteria were the presence of neurologic, psyichiatric or pulmonary diseases, severe hearing loss, previous demolitive surgery (glossectomy, total pharyngectomy, floor of mouth resection or mandibulectomy) and chemo or radiotherapy treatment for a recurrent or second tumor. The enrolled patients were divided into four groups, depending on the type of substitution voice used, namely, the tracheoesophageal speakers group (TES), the esophageal speakers (ES) group, the electrolarynx users (EL) group, the no voice patients (NV) group. In Table 1, the clinical and sociodemographic data are presented.

### 2.2. Quality of Life Measurements 

A phoniatric evaluation was performed at the beginning of the examination (T0), by administering to the patients two assessment questionnaires: the first one was the “*Voice-Related Quality of Life (V-RQoL)*”, which investigates QoL regarding the use of the voice [23], whereas the second questionnaire is “Self-Evaluation of Communication Experiences after Laryngeal Cancer (SECEL)”, which evaluates the subjective perception the patient has of his own vocal emission [23]. The purpose of the questionnaires was accurately explained before the administration. Patients who needed further help in answering the various items were assisted by speech therapists.

#### 2.2.1. The V-RQoL Questionnaire (Appendix A)

The V-RQoL questionnaire (Appendix A) is a self-assessment asset that evaluates the socioemotional, physical and functional aspects of the patient in relation to his voice disorder and makes it possible to get a lot of valuable information regarding the patient’s quality of life [24,25,26,27,28]. The version of the V-RQoL questionnaire used in this study is divided into 2 sections. The first section includes 10 items that investigate the difficulties encountered in daily life, regarding both functional and social aspects. The section is divided in two domains—physical and mental functioning. The physical functioning domain consists of six questions (items 1–3, 6, 7, 9) investigating the possible problems encountered during phonation, as the need to frequently get air, or other difficulties impairing or reducing the patient’s daily activities. The four questions for mental functioning (items 4, 5, 8, 10) reflect the social and emotional consequences related to vocal problems, such as restrictions on interpersonal relationships, perception of depressive symptoms or associated anxiety. To each item, a score from 1 to 5 is assigned, depending on the degree of difficulty perceived for the situations described in each case (1: no problems; 2: some; 3: moderate; 4: marked; 5: severe). The various score ranges are associated to a judgement for the QoL are: 10–15: excellent; 16–20: very good; 21–25: good; 26–30: quite good; above 30: low. A proper algorithm resizes the scores in a range from 0 to 100: high partial and/or total scores indicates a worse quality of life, since the lower is the score, the lower the impact that voice alteration has on the patient’s life. The second section evaluates the patient’s vocal outcomes and consists of 5 items dealing with the patient’s own judgment about the substitution voice (the first item, hereafter referred to as Question A, namely “*How would you generally define your speaking voice is?*”); the potentially reduced ability to be understood in a noisy environment; the possible interferences on social or working activities; the presence of swallowing problems; and the patient’s judgement of the entity of his efforts when speaking.

#### 2.2.2. The SECEL Questionnaire (Appendix B)

The SECEL, a reliable and validated self-evaluation questionnaire of communicative experience after laryngectomy surgery [29,30,31], measures the communicative impairment in the laryngectomees and the effects of the voice therapy and rehabilitation on daily life activities, independent of the rehabilitative method to which they were submitted. The questionnaire is composed of 35 items, of which the first 34 are grouped into 3 subscales: general (5 items), environment (14 items) and attitude (15 items). A score from 0 (never) to 3 (always) is assigned to each item, with a score range of 0–102 for the total score, 0–15 for the general subscale, 0–42 for the environment subscale and 0–45 for the attitude subscale. The higher the score, the greater the perception of communication dysfunction.

The final item 35 deals with the impact of TL in reducing the desire to communicate: “*Do you talk the same amount now as before your laryngectomy?*”, and the answer can be “YES”, “MORE” or “LESS”.

### 2.3. The INFVo Scale

In this scale five parameters are defined:

I → overall impression, which reflects the overall vocal quality, a parameter indicating the combined impression caused by all the properties of the voice, such as pleasant/unpleasant to listen to, fluent or cut, good volume or not, intelligible or not, etc.;

N → unintended additive noise, which reflects the amount of annoyance caused by the audibility of all types of uncontrolled noises, such as breathy or bubbly noises, clicks and any other sounds produced during speech;

F → fluency, which reflects the perceived smoothness and fluidity of the sound production and remarkably depends on the patient’s ability to manage the expiratory air; samples containing a lot of hesitations between successive sounds and within continuant sounds (for example vowels and some consonants) score badly;

Vo → quality of voice, which means that the produced speech voicing is supposed to be voiced or unvoiced; voices that produce a lot of breathy noise and contain little or no voiced segments get a low score.

All voice recordings were used for the auditory perceptual evaluation performed on the INFVo scale by a judge expert team composed of four people blinded to the study speech and language pathologists with more than 20 years of experience in laryngectomees treatment and rehabilitation. They all worked in different clinical departments. All referees were previously trained with the INFVo scale using samples of male and female substitution voices that were not included in this study. Each judge generally listened to the recording of the speech sample of the 89 subjects in a randomized order, with a 15 min rest interval every 20 patients. A score from 0 to 10, presented on a visual-analogic scale (VAS) and orientated from the left (lowest score) to the right (highest score), was assigned for each parameter of the scale: the higher is the score, the worse the perception of voice quality. To each parameter, a score ranging from 0 (very good substitution voice) to 10 (very deviant substitution voice) was attributed. The same auditory perceptual evaluation was repeated after 15 days.

#### Speech Samples

The audio samples of the enrolled patients were recorded by means of a Rode NT1 condenser microphone placed at a distance of about 10 cm from the patient’s mouth and with an inclination of about 45°, in order to reduce the air flow effect. The free computer software package for speech analysis in phonetics PRAAT, an interactive model for manually segmenting recordings into vocal expressions proposed by Boersma and Weenink in 1996 [32], was used. Originally, the samples were recorded with a sampling frequency of 32 kHz, and they were subsequently downsampled again to 16 kHz, stored as files in WAV format on a personal computer [33], then analyzed by the auditory model of Van Immerseel and Martens [34]. All recordings were made in a silent quiet room, with an environment noise > 50 dB. The vocal sample included:Emission of the isolated and prolonged vowel /*a*/ at a comfortable pitch and intensity;Automatic seriations, as progressive numbering from 1 to 10;reading aloud a short, standardized text, consisting of 5 sentences for a total of 100 syllables;A repetition test (the examiner says a word and the patient must repeat it) of phonetically balanced words, which includes all the phonemes of the Italian language;An oral diadochokinesis test, in which the patient was asked to pronounce the 3 syllables of the sequence [PA/TA/KA] in five seconds as quickly as possible.

The following parameters were recorded:
Maximum phonation time (MPT), in seconds, by measuring the length of the prolonged emission;Proportion of the vocal structure (PV), which depends on the number of speech pauses during the production of automatic series;Percentage of speech (in syllables per second), which is calculated on the basis of the time needed to read the chosen text;Number of phonetic distortions or substitutions occurred during the word repetition test;Oral diadochokinesis (DDK), in syllables/second, through the rapid reproduction of a syllabic sequence;Presence/absence of noise during speech.

### 2.4. Statistical Analysis

Demographic and clinical data and other disease-related variables were statistically described with the use of frequencies (percentages when appropriate) for categorical variables, and the mean, SD and 95% Confidence Interval (CI), for quantitative variables. The cross-tabulation analysis, based on Person’s chi-square, was performed to test the difference in any demographics between groups.

Intra-judge and inter-judge reliability were verified by calculating Cohen’s K coefficient. For the perceptual assessments and after excluding one of the four experts due to lower intra-evaluator reliability, overall Cohen’s K was set to 0.80.

In order to correlate the patient’s perception of its voice with its impact on his quality of life, each of the scales administered was analyzed by using the correlation coefficient *r* Pearson between each pair of variables. An ANOVA test was performed to compare the mean values obtained by the administration of the INFVo, SECEL and VRQoL scales to the four different groups of patients (TES, ES, ELS and NV) under investigation. Bonferroni’s test was used for multiple comparisons of the difference of means.

In order to detect which item/variable or which mix of them significantly diversified the 4 groups, a discriminant analysis (hereafter, DA) was carried out. DA was performed by entering all variables and by selecting, through a stepwise method, the best set of discriminating variables. The criterion for controlling the stepwise selection was the maximum Wilks’ lambda. The mathematical objective of DA is to weight and linearly combine the discriminating variables in some fashion so that the four groups (or clusters) are forced to be as statistically distinct as possible. The statistical theory of DA assumes that the discriminating variables have a multivariate normal distribution, and that they have equal variance–covariance matrices within each group. In practice, the technique is very robust and these assumptions do not need to be strongly adhered to. The discriminant scores were derived by maximizing the quadratic distance of Mahalanobis from the centroid of the four groups [35]. All *p*-values were two-sided and values less than 0.05 were considered significant. Statistical analysis was performed with the software IBM SPSS Statistics, v.20.0 (IBM Corp. Armonk, NY, USA).

The study is in accordance with relevant guidelines and regulations. The study was approved by the institutional review board committee of Federico II University of Naples, Naples, Italy (2021/206584). Informed written consent from the patients was obtained.

## 3. Results

No significant difference was found in any demographics between groups (Table 1, last column, χ2 Sig.).

### 3.1. V-RQoL Questionnaire

As resulting from the V-RQoL questionnaire, TES patients claimed to have a better quality of life than ES and NV patients. Scores attesting an excellent or good QoL were obtained in the 81% of TES patients, whereas only five subjects of this group (9%) got a total score greater than 30. ES laryngectomees achieved values ranging from “bad” (43%) to “very good” (28%), whereas the majority of NV patients (89%) were very unsatisfied, mainly because of their vocal impairment. All ELS patients judged positively their post laryngectomy condition. The results are reported in Figure 1.

Concerning the judgement of their voices (Question A) the majority of TES patients (93%) were satisfied with their substitution speech (adequate in 59%, good in 30% and excellent in 4%) compared to only 57% of the ES patients (adequate in 43% and good in 14%). All the ELS patients positively judged their means of verbal communication, while 67% of NV patients considered their whispered new voice very unsatisfactory. The remaining laryngectomees (33%) were completely voiceless. The statistical data regarding the judgement of the patients on Question A are reported in Figure 2.

### 3.2. SECEL Questionnaire

In the TES group, only two patients (4%) scored below the cut off, while the limit of the average was reached by two more subjects (4%). The environment domain is much more compromised, especially for the NV (mean = 32.89) and ES (mean = 20.14) patients, wherever slightly higher values are reported in the general and attitude subscales for all the four categories of speakers.

No significant group differences were found between the mean of TES, ES and NV patients (ANOVA, F between groups 2.22, Sig. 0.09).

Regarding item n.35, an adaptation to the new substitution voice method suitable to favor an optimal recovery of the communicative function was observed only in 14 patients (16%), even though the social habits were not influenced by TL in 31% of cases. More precisely, 24 of TES patients (44%) tend to use the voice function less to communicate, while 8 ES patients (62%) reduced their verbal interactions and 2 ELS (50%) and 13 NV patients (78%) confirmed the remarkable impact that their speech difficulties had on their communication. The results obtained from the SECEL scale for each group of laryngectomees are reported in Figure 3 and Figure 4.

In Figure 4, data concerning the item 35 are reported.

### 3.3. INFVo Scale

Data obtained from the INFVo scale for each group of laryngectomees are reported in Figure 5.

Good intelligibility scores for the alaryngeal voice were obtained in all the groups of patients analyzed. ES and ELS laryngectomees generally got higher scores for the items evaluating the presence of noise, fluency of speech and quality of the voice. In fact, TES patients scored an average of 2.5 points (range 0–7 points, SD = 1.9) for impression, 2.9 points (range 0–7 points,) for noise, 2.2 points (range 2–7 points, SD = 1.9) for fluency and 2.5 points (range 0–6 points, SD = 1.9) for quality of voice. ES patients totaled 6.0 points (range 2–9 points, SD = 2.3) for impression, 7.2 points (range 4–10 points, SD = 2.0) for noise, 6.5 points (range 3–10 points, SD = 2.4) for fluency and 5.0 points (range 3–7 points, SD = 1.3) for quality of voice. ELS patients obtained an average of 2.0 points (range 2–2 points, SD = 0.0) for impression, 5.5 (range 4–7 points, SD = 1.7) for noise, 2.5 points (range 2–3 points, SD = 0.6) for fluency and 6.0 points (range 5–7 points, SD = 1.1) for quality of voice. The group of NV patients got 9.1 points (range 7–10 points, SD = 1.0) for impression, 9.1 points (range 8–10 points, SD = 0.8) for noise, 9.4 points (range 8–10 points, SD = 0.9) for fluency and 8.2 points (range 3–10 points, SD = 2.3) for quality of voice.

Significant ANOVA differences between groups were found: I (F = 76.5, Sig. < 0.001), N (F = 67.5, Sig. < 0.001), and Vo (F = 44.38, Sig. < 0.001).

### 3.4. Discriminant Analysis

The method based on the DA is a multifactorial statistical solution. Using the stepwise model, after each step in which a variable is added, all candidate variables in the model are checked to see if their significance is reduced below the specified tolerance level (see below). If a nonsignificant variable is found, it is removed from the model. This way, variables showing excessive variability, for instance, “scoring errors due to the limitations associated with observation” are excluded from the statistical linear regression, namely the variable F (fluency).

The variables included in the stepwise analysis and the corresponding Wilks’ lambda are reported in the Table 2. Table 2 reports information necessary for judging how many discriminant functions should be derived. The eigenvalues and their associated canonical correlations denote the relative ability of each function to separate the groups. Clearly, the third function is useless (STEP 3). The right side of Table 2 shows the change in Wilks’ lambda as the information in successive discriminant functions is removed. The larger the lambda, the less discriminating power is present. The stepwise model selects independent variables for entry into the analysis on the basis of their discriminating power. The criterion is overall multivariate *F* ratio for the test of differences among the group centroids. The variable, which maximizes the *F* ratio, also minimizes Wilks’ lambda, a measure of group discrimination. An additional test is performed before a variable is actually accepted. This is a test to see if the tolerance for this variable is sufficiently high. An extremely low tolerance level is a sign that the program would have difficulty inverting a covariance matrix, which included this variable.

The eigenvalues, the percent of explained variance and canonical correlation are reported in Table 3, where the first three canonical discriminant functions were used in the analysis.

Further evidence regarding the group differences can be derived from the plot of group centroids (Figure 6). These are the mean discriminant scores for each group on the respective functions. The centroids summarize the group locations in the space defined by the discriminant functions. Three distinct groups, namely TES, ES and NV, can be seen by means of the first function (i.e., by the variable I, impression). Two groups, namely ELS and the cluster formed by TES, ES and NV, can be seen by the second function (i.e., by the variable N, noise). The centroids are clearly separated confirming that the discrimination is statistically significant.

Figure 6 shows the plot of centroids of the four groups for the first two canonical discriminant functions obtained by the DA.

Another use of the classification is in testing the adequacy of the derived discriminant functions. By classifying the cases used to derive the functions in the first place and comparing predicted group membership with actual group membership, one can empirically measure the success in discrimination by observing the proportion of correct classification. The classification results are reported in Table 4. We observe that the 89% of the TES group was correctly classified. The ES, ELS and NV groups were classified at 69%, 100% and 76% respectively. In total, the 84% of original grouped cases were correctly classified (Table 4).

The results of multiple comparisons Bonferroni’s test of mean difference of Question A, and INFVo (i.e., I, N and Vo), namely the variables selected by the DA, among groups, are reported in the Table 5. Note that the variable F is not included because it is excluded by the DA.

Table 6 shows the correlation matrix: the patient’s self-report (Question A) with the INFVo scale, SECEL and VRQoL questionnaires. The Pearson product–moment correlation coefficient is a measure of the strength and direction of association that exists between two variables measured on at least an interval scale. The higher the value, the stronger the correlation between the variables.

## 4. Discussion

Vocal impairment subsequent to total laryngectomy seriously affects the patient’s communicative ability, leading to social and psychological disabilities that remarkably influence the patient’s health and quality of life [36,37,38]. This is widely reported in the literature [39,40,41,42,43,44,45,46,47,48,49,50] on the improvement in vocal and life quality in patients rehabilitated with tracheoesophageal prostheses. Our results confirmed these data, since TES patients obtained low values on the VAS scale. However, the results obtained demonstrated a general decrease of the quality of life affecting all four groups; mainly, as easily supposed, the NV patients, whose communicative difficulties are evident in the personal judgement of their own voice, defined as inadequate or absent in 67% and 33% of the patients, respectively.

As resulting from the V-RQoL scale, TES patients claimed to have a better quality of life (mean 24.8) than ES (mean 27.6) and NV patients (mean 30.6). Concerning *Question A,* TE patients showed the best satisfaction regarding their substitution voice (mean 2.7), followed by ES (2.9) and NV (4.0) laryngectomees. The answers of the ELS group were not analyzed due the small sample size. However, the mean values obtained are not statistically significant. Our data are similar with those previously reported in the literature by some AA [51,52], which stated that none of the methods of speech rehabilitation studies achieved a significant difference regarding satisfactory outcomes for self-reported vocal function, even though TES patients reported improved voice performance.

The SECEL scale quantifies the level of adaptation of patients to the new substitution voice and investigate the general, environmental and social conditions that mostly invalidated oral communication. The environment subscale was the most compromised area. In fact, the main problem the patients faced, regardless of the phonatory method used, involves conversation in a noisy environment, especially when reverberation is present. To a lesser extent, a considerable score was also obtained for the general subscale, because of the feeling of discomfort experienced by patients in a dialogue situation and the perception that the quality and fluency of the voice tend to improve during the conversation, when the patient is better accustomed to the new situation. Moderate scores were attributed to the attitude subscale, underlining the difficulties encountered in social interactions. The overall higher scores were achieved by ES (mean 32.2) and NV (mean 40.9) patients, showing this last group faced greater communication problems. However, statistical analyses of the data obtained are not significant even in this case.

The accurate evaluation of the changes in the communication methods adopted by laryngectomees and the perceived level of adaptation to the substitution voice make the well-adapted patients more easily identifiable from those who refused the new way of communication. To identify the patients who need proper rehabilitation, a specific cut-off value is recommended. The score for an adapted patient should be 36, with an SD of 12 points; if the score is greater than 60, it is advisable to refer the patient to specific counseling in order to find a coping strategy for better adaptation to the new clinical-functional condition 30. In our experience, the SECEL has proven a useful screening tool, because the rehabilitative and psychosocial needs of the unsatisfactory patients are well investigated, making it possible to establish a proper rehabilitative strategy to face their communicative handicap, especially if psychological support should be included in the rehabilitative program for better adaptation to the new substitution voice [44,45].

As part of the evaluation methods of the voice in the laryngectomized patient, the INFVo scale has proven to be a useful tool for the description and analysis of the psychoacoustic characteristics of the vocal signal. TES patients got the lowest scores, which means they benefitted from the best psychoacoustic voice outcomes, whereas ES patients got higher scores for the presence of parasitic noises and reduced vocal autonomy due to several interruptions for air supply. The most likely explanation for this is related to the pulmonary driven type of TE speech (tidal volume of 500–600 mL), which allows for a more stable and controlled airflow. Only a minimal volume of air (about 60–80 mL) is available for ES patients, therefore, the uneasy difficult air pressure control leads to a shorter phonation time. Similar higher scores were obtained in NV patients for the noises produced by the excessive effort involved in the attempt to phonate and for the marked unintelligibility related to the total absence of voice.

Based on the DA, it is notable that the most significant parameter between mean differences and discriminatory power among the four groups was the variable “I” (overall impression), since the mean varies from 2.5 (TES), to 5.9 (ES) and 9.1 (NV), considering that the higher the score, the worse the overall impression of the voice. The second significant parameter was the variable “N”, whose means are 2.9 for TES patients, 7.2 for ES patients, 5.5 for ELS patients and 9.1 for NV patients. Finally, the third parameter was Vo: 2.5 (TES), 5.1 (ES), 6.0 (ELS) and 8.2 (NV). The “F” parameter is strictly related to the “I” and “N”—and these are the reasons why is not considered in the DA.

In our experience and referring to the statistical analysis carried out, the INFVo scale has proven a valid investigative instrument for assigning the patient’s substitution voice to a proper group. In fact, from the discriminant analysis, around 84% of the laryngectomees are correctly classified as part of their group, thus demonstrating the reliability of the INFVo scale. It is also notable that the judgment of the patients on their own voice and those of the referees show highly significant correlations.

As was clearly predictable, what has been shown to be statistically significant, independent of the type of substitution voice achieved, is the frequency in the speech therapy sessions attended (Table 1 Sig. 0.002).

## 5. Conclusions

Although speech rehabilitation for the acquisition of a substitution voice offers a new way of communication for the laryngectomized patients, their satisfaction is closely related both to their own voice perception and to the impression and judgment of the people they interact with. TES seems to offer the best vocal performances compared to the other rehabilitative methods; nevertheless, the quality of life is decreased to an extent in every group.

This evidence supports the statement that good speech intelligibility may not correspond to patient’ good judgement about their outcome and their own QoL perception. The frequently cited tendency to social withdrawal, especially in unusual social conditions, could be related to this aspect.

Data obtained from the V-RQoL and SECEL questionnaires also demonstrated that no significant differences exist among the four groups. Improving the patient’s adaptation to the new phonatory condition is mandatory. This goal can be primarily achieved planning pre-operative counselling for explaining the postoperative anatomo-functional condition and the need for rehabilitative procedures. Repeated meetings with laryngectomees already rehabilitated, possibly of same age, gender and socio-cultural asset, in order to exchange experiences and sensations, are strongly advisable. This way, better acceptance and adaptation to the new communication method are easier to achieve, thus promoting social reintegration and, as consequence, improving the patient’s quality of life.

The INFVo scale has proven a valid tool for the instrumental analysis of the substitution voice for several reasons: mainly the remarkable accuracy in classifying the single patient to the proper vocal group and the high significance between patients and expert team’s judgement.

## Figures and Tables

**Figure 1 jpm-13-00570-f001:**
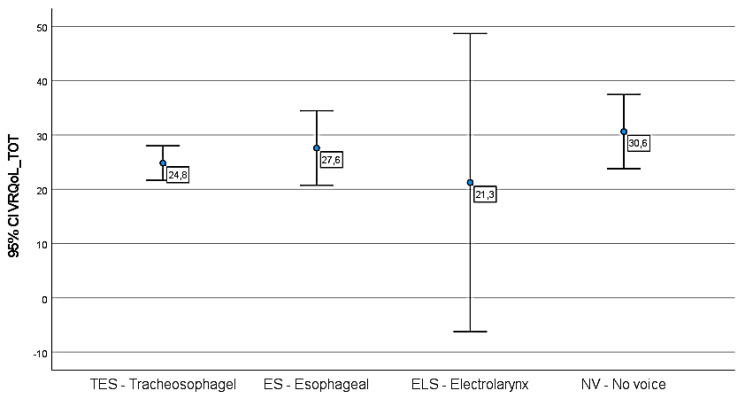
*VRQoL self*-assessment, circle represents the mean, bars represent 95% CIs, labels report the mean. Score range: 10–15 excellent, 16–20 very good, 21–25 good, 26–30 quite good, above 30 low.

**Figure 2 jpm-13-00570-f002:**
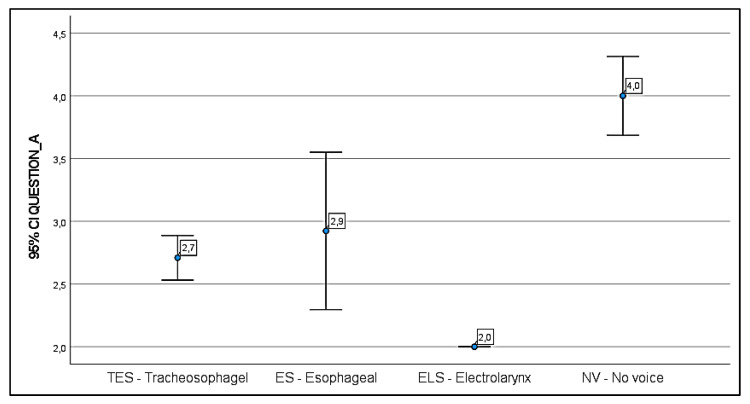
Question A: circle represents the mean, bars represent 95% CIs, labels report the mean. Score legend: 1: no problem; 2: some; 3: moderate; 4: marked; 5: severe.

**Figure 3 jpm-13-00570-f003:**
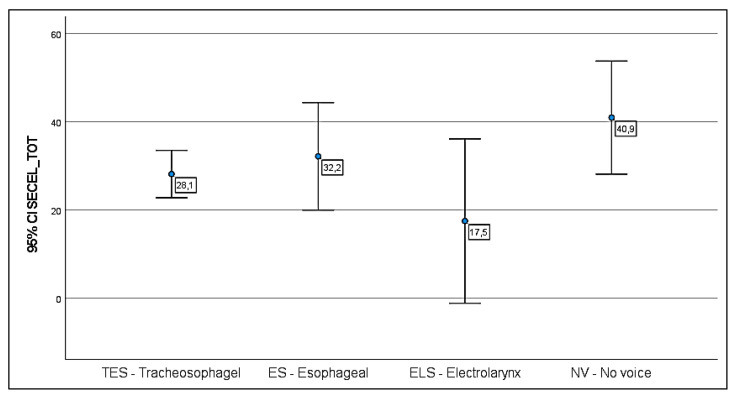
SECEL self-assessment, circle represents the mean, bars represent 95% CIs, labels report the mean. Score legend: cut-off value 36.

**Figure 4 jpm-13-00570-f004:**
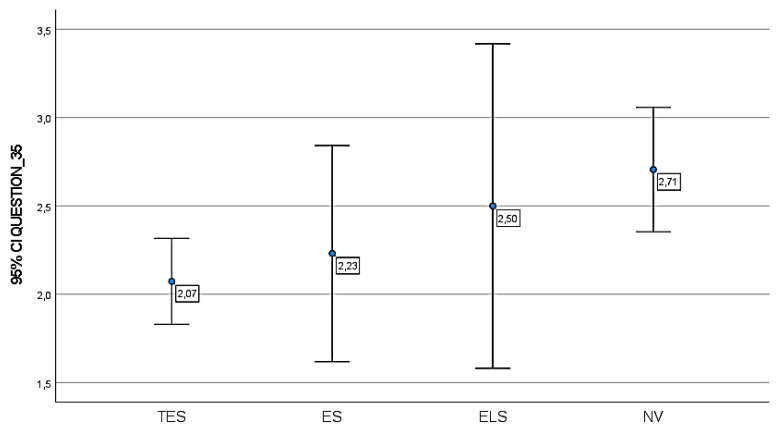
Item n.35 “*Do you talk the same amount now as before your laryngectomy?*”—self-assessment, circle represents the mean, bars represent 95% CIs, labels report the mean. Score legend: cut-off value 36 Score legend: 1 = YES; 2 = MORE, 3 = LESS. No significant ANOVA difference between groups was found (F = 2.414, Sig. = 0.72).

**Figure 5 jpm-13-00570-f005:**
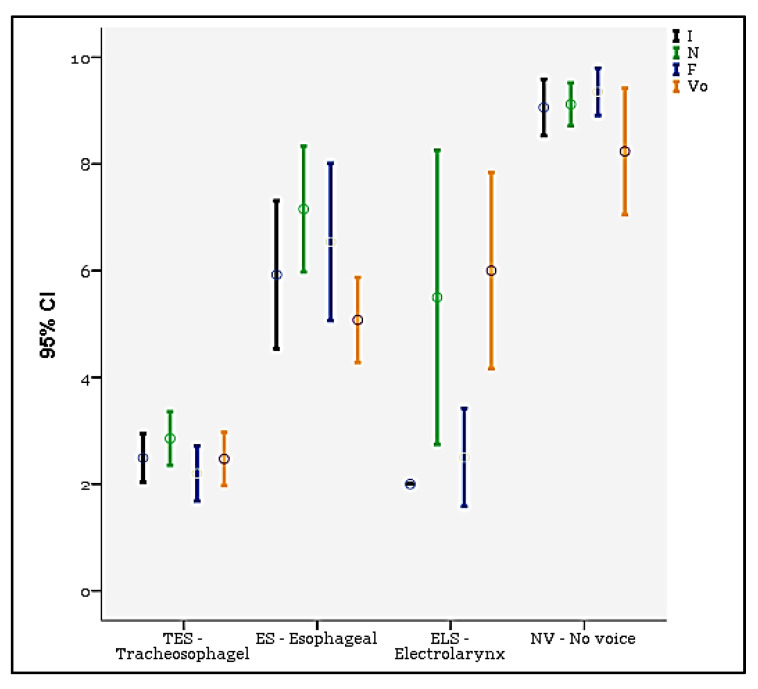
INFVo judgment of specialists on alaryngeal speech, circle represents the mean, bars represent 95% CIs. Legend: I, impression; N, noise; F, fluency; Vo, quality of voice. Score legend: 0 (very good substitution voice)–10 (very deviant substitution voice).

**Figure 6 jpm-13-00570-f006:**
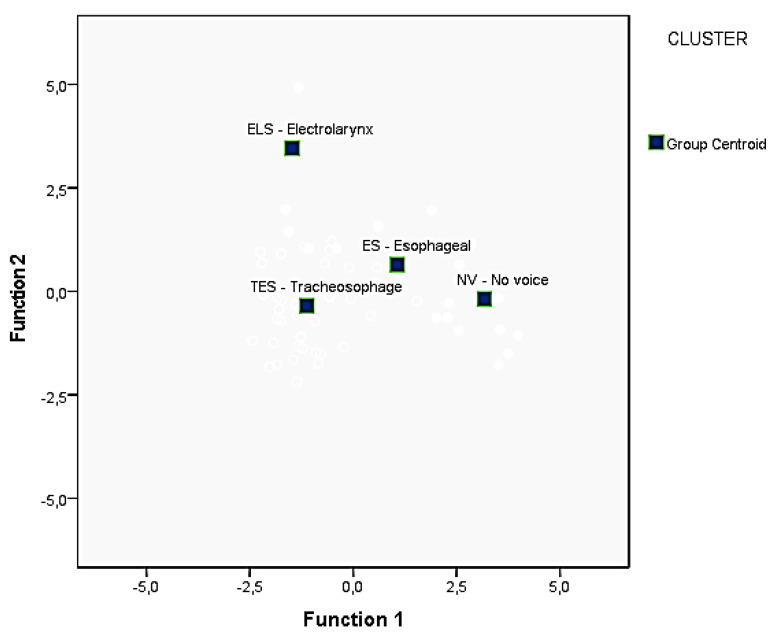
Plot of centroids of the groups.

**Table 1 jpm-13-00570-t001:** Clinical and sociodemographic data.

Characteristics	TES	ES	ELS	NV	$\chi^{2}\;\mathrm{Sig}.$
Sample size	55 (62%)	13 (15%)	4 (4%)	17 (19%)	
Age mean (range)	64 (45–78)	65 (44–81)	63 (57–69)	62 (49–76)	n.s.
Male	45 (82%)	12 (92%)	4 (100%)	16 (94%)	0.750
Female	10 (18%)	1 (8%)	0 (0%)	1 (6%)	
Non-smoker	10 (18%)	0 (0%)	0 (0%)	1 (6%)	0.460
Former smoker	40 (73%)	13 (100%)	4 (100%)	16 (94%)	
Smoker	5 (9%)	0 (0%)	0 (0%)	0 (0%)	
Time since LT (mean, ys)	2.80	2.38	3.00	2.12	
Radiotherapy	32 (58%)	6 (46%)	2 (50%)	10 (59%)	0.062
Chemotherapy	14 (25.5%)	4 (30.8%)	0 (0%)	4 (23.5%)	0.760
Speech therapy	42 (76.4%)	10 (77%)	2 (50%)	4 (23.5%)	0.002

**Table 2 jpm-13-00570-t002:** Stepwise model of discriminant analysis: variables in the analysis.

STEP		TOLERANCE	F TO REMOVE	WILKS’ LAMBDA
1	I	1.000	76,494	
2	I	0.307	20,822	0.296
	N	0.307	16,654	0.270
3	I	0.286	22,510	0.248
	N	0.259	10,620	0.189
	V_0_	0.404	6587	0.169

**Table 3 jpm-13-00570-t003:** Main parameters of DA.

Function	Eigenvalue	% of Variance	Cumulative %	Canonical Correlation
1	3.104 ^a^	77.1	77.1	0.870
2	0.711 ^a^	17.7	94.8	0.645
3	0.211 ^a^	5.2	100.0	0.417

a: First 3 canonical discriminant functions were used in the analysis. The model of the first function is: Question A × 0.337 + I × 0.766 + N × 0.239 − Vo × 0.082. The model of the second function is: Question A × (−0.288) − I × 1.476 + N × 1.386 + Vo × 0.498, where the values are standardized canonical discriminant function coefficients.

**Table 4 jpm-13-00570-t004:** Classification results.

Group			Predicted Group Membership				
			TES-Tracheoesophagel	ES–Esophagel	ELS–Electrolarynx	NV–No Voice	Total
**Original**	**Count**	TES-Tracheoesophagel	49	5	0	1	55
		ES–Esophagel	2	9	0	2	13
		ELS–Electrolarynx	0	0	4	0	4
		NV–No Voice	0	4	0	13	17
	%	TES-Tracheoesophagel	89.1	9.1	0.0	1.8	100.0
		ES–Esophagel	15.4	69.2	0.0	15.4	100.0
		ELS–Electrolarynx	0.0	0.0	100.0	0.0	100.0
		NV–No Voice	0.0	23.5	0.0	76.5	100.0

84.3 % of original grouped cases correctly classified.

**Table 5 jpm-13-00570-t005:** Multiple comparisons of Bonferroni’s test of mean difference of Question A, I, N and Vo.

Dependent Variable	Group Comparison	Mean Difference	Sig.
Question A	TES vs. NV	−1.291	<0.0001
	ES vs. NV	−1.077	<0.0001
	ELS vs. NV	−2.000	<0.0001
I	TES vs. ES	−3.432	<0.0001
	TES vs. NV	−6.568	<0.0001
	ES vs. ELS	3.923	<0.0001
	ES vs. NV	−3.136	<0.0001
	ELS vs. NV	−7.079	<0.0001
N	TES vs. ES	−4.299	<0.0001
	TES vs. ELS	−2.645	0.023
	TES vs. NV	−6.263	<0.0001
	ES vs. NV	−1.964	0.016
	ELS vs. NV	−3.618	0.002
Vo	TES vs. ES	−2.604	<0.0001
	TES vs. ELS	−3.527	0.003
	TES vs. NV	−5.763	<00001
	ES vs. NV	−3.158	<00001

**Table 6 jpm-13-00570-t006:** Correlation matrix.

		QUESTION A	I	N	F	Vo	SECEL_TOT	VRQoL_TOT
QUESTION A	Pearson Correlation	1	0.520 **	0.446 **	0.505 **	0.403 **	0.462 **	0.334 **
	Sig. (2-tailed)		<0.001	<0.001	<0.001	<0.001	<0.001	<0.001
	N	89	89	89	89	89	89	89
I	Pearson Correlation	0.520 **	1	0.925 **	0.968 **	0.857 **	0.281 **	0.234 *
	Sig. (2-tailed)	<0.001		<0.001	<0.001	<0.001	0.008	0.27
	N	89	89	89	89	89	89	89
N	Pearson Correlation	0.446 **	0.925 **	1	0.953 **	0.894 **	0.288 **	0.201 *
	Sig. (2-tailed)	<0.001	<0.001	<0.001		<0.001	0.006	0.59
	N	89	89	89	89	89	89	89
F	Pearson Correlation	0.505 **	0.968 **	0.953 **	1	0.878 **	0.307 **	0.246 *
	Sig. (2-tailed)	<0.001	<0.001	<0.001		<0.001	0.003	0.020
	N	89	89	89	89	89	89	89
Vo	Pearson Correlation	0.403 **	0.857 **	0.894 **	0.878 **	1	0.290 **	0.207
	Sig. (2-tailed)	<0.001	<0.001	<0.001	<0.001		0.006	0.052
	N	89	89	89	89	89	89	89
SECEL_TOT	Pearson Correlation	0.462 **	0.281 **	0.288 **	0.307 **	0.290 **	1	0.577 **
	Sig. (2-tailed)	<0.001	0.008	0.006	0.003	0.006		<0.001
	N	89	89	89	89	89	89	89
VRQoL_TOT	Pearson Correlation	0.334 **	0.234 **	0.201	0.246 *	0.207	0.577 **	1
	Sig. (2-tailed)	0.001	0.027	0.059	0.020	0.052	<0.001	
	N	89	89	89	89	89	89	89

Legend: ** Correlation is significant at the 0.01 level (2-tailed). * Correlation is significant at the 0.05 level (2-tailed).

## Data Availability

Data are available upon reasonable request.

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
