# Peer review of "Subjective Perception and Psychoacoustic Aspects of the Laryngectomee Voice: The Impact on Quality of Life"

_jpm, 2023, doi:10.3390/jpm13030570_

Round 1

Reviewer 1 Report

Thank you for the opportunity to review the article entitled “Subjective perception and psychoacoustic aspects of the laryngectomee voice: the impact on quality of life”.

This study aims to evaluate the impact of vocal impairment on the quality of life of laryngectomees. The results are considered useful for the selection of substitution voice in laryngectomees. Further studies pertaining to the improvement of the patient's adaptation to the new phonatory condition are expected.

- There are many typos in the text and tables(e.g., line 80, table 7, …).

- The resolution of the figures is not high enough.

- The questionnaire could be provided as an appendix.

- Laryngectomized patients with preoperative good vocal quality tend to report greater vocal impairment and lowered quality of life right after laryngectomy. They also show a tendency to adapt to newly established voice as they get familiar with the new voice. Therefore, detailed information on when the postoperative voice evaluation was performed should be presented in Table 1.

- Please clarify what tables 5 and 6 represent.

- Due to the nature of laryngeal cancer, most of the participants were male. Nevertheless, the F0 of most alternative phonation tends to be low, so there may have been a difference in voice satisfaction according to gender. If there was a gender difference in the current data, it would be good to add a discussion about it.

- Although this study was retrospective in nature, was informed consent obtained from all participants, or was it waived?

Author Response

REVISORE 1

thank you for the opportunity to review the article entitled “Subjective perception and psychoacoustic aspects of the laryngectomee voice: the impact on quality of life”.

This study aims to evaluate the impact of vocal impairment on the quality of life of laryngectomees. The results are considered useful for the selection of substitution voice in laryngectomees. Further studies pertaining to the improvement of the patient's adaptation to the new phonatory condition are expected.

- There are many typos in the text and tables (e.g., line 80, table 7, …).

- The resolution of the figures is not high enough.

- The questionnaire could be provided as an appendix. OK

- Laryngectomized patients with preoperative good vocal quality tend to report greater vocal impairment and lowered quality of life right after laryngectomy. They also show a tendency to adapt to newly established voice as they get familiar with the new voice. Therefore, detailed information on when the postoperative voice evaluation was performed should be presented in Table 1. INSERTED MEAN YEARS. Our patients underwent in the previous 3 years; the vocal evaluation was done at the same time for all.

- Please clarify what tables 5 and 6 represent. THE INFORMATION WAS ALREADY DONE ON PAG.13. (now Tables 3 and 4)

- Due to the nature of laryngeal cancer, most of the participants were male. Nevertheless, the F0 of most alternative phonation tends to be low, so there may have been a difference in voice satisfaction according to gender. If there was a gender difference in the current data, it would be good to add a discussion about it.

NOT INCLUDED IN THE RESULTS BECAUSE NO RELEVANT STATISTICAL DIFFERENCES WERE REPORTED

- Although this study was retrospective in nature, was informed consent obtained from all participants, or was it waived? OBTAINED FROM ALL PATIENTS

Reviewer 2 Report

P1 L13 Abbreviation first used in the text

P1 L15 Abbreviation first used in the text

P1 L26 This is better suited for discussion, as your team searched for correlations between the patient's different psycho-perceptual parameters and vocal function perception.

P1 L33 But it is not the only approach. Please refer to other treatment modalities as well. I suggest using NCCN guidelines for the reference.

P1 L45 This is a profound statement. A reference would be advisable.

P2 L47 The sentence is too long and hard to follow. Please consider splitting it into two.

P2 L49 More commonly abbreviated as tracheoesophageal prosthesis  (TEP). I advise changing and using consistently throughout the text.

P2 L50 Please revise:

TEP prevents saliva or food from entering the lower respiratory tract and enables air to enter the esophagus, allowing the pharyngoesophageal (PE) segment to vibrate.

P2 L57 The air is voluntarily sucked in and not administered. Please revise.

P2 L58 Vibrations.

P2 L59 As described in your reference, the electrolarynx produces vibrations but not sound. Please revise the sentence not to confuse the reader.

P2 L62 What does this abbreviation refer to?

P2 L66 Did you mean substitution voice?

P2 L67 Could you provide a reference?

P2 L73 Use the name of the scale first and the abbreviation afterward. Please keep consistency throughout the text.

P2 L77 Are you writing about voice or speech? Please correct accordingly.

P2 L92 Please revise from 44 to 81 years

P2 L92 Some of the patients might have had both, and it is known that combined treatment modalities have a more significant impact on QoL. Please elaborate

P2 L94 Were more patients included in the study, and where these patients excluded or not recruited for this study? Please edit the sentence to make it clearer. Also, did you consider not including patients with neurodegenerative conditions (e.g., hearing loss and poor motor function) – disclose this information either way.

P3 L99 the name first and the abbreviation afterward. Please keep consistency throughout the text.

Table 1 mentions 75 males, but 77 are reported. Please revise.

Table 1 You mention 14 females, but only 12 are reported. Please revise.

Table 1 You do not mention smoking or XRT anywhere further in the text. Please consider if this data is relevant to the reader in the context of this article.

P3 L109 Was the explanation provided before administering the questionnaire? Please mention either way.

P3 L112 The full name is used at the first mention, and then the abbreviation is used throughout the text. Please revise and keep consistency throughout the text.

P3 L112 Questionnaires are typically presented as supplementary material. Please reconsider.

P4 L130 Quality of life?

P4 L136 You mentioned that the second section evaluates the patient's vocal outcomes. Swallowing is not one of them. Please clarify.

P5 L130 Overexplaining, especially with the questionnaire provided. Please reduce or consider removing this part.

P5 L140 Is it a questionnaire or a scale? Please revise and use consistently throughout the text.

P5 L140 Questionnaires are typically presented as supplementary material. Please reconsider.

P5 L147 Is it a questionnaire or a scale? Please revise and use consistently throughout the text.

P5 L109 This is better suited for introduction or discussion. Please consider moving.

P6 L150 of

P6 L154 This is better suited for introduction or discussion. Please consider removing it.

P8 L1161-176 The paragraphs should rarely consist of a single long sentence. Please consider shortening the explanation and providing one consistent paragraph.

P8 L177 Were do you  recording speech or voice or both?

P8 L178 You recorded both speech and phonation, 'The audio samples' would be more appropriate

P8 L178 It would be helpful to know the exact model of the microphone if other researchers try to replicate your research.

P8 L186 This is not very silent. Please revise.

P8 L188 The IINFVo scale evaluates speech. Why did you need to record phonation? Please explain or remove it.

P8 L190 It would be helpful to know the text in your native language and translated if other researchers try to replicate your research.

P8 L192-207 You never mention this again in the text or use it with the IINFVo scale. How is this relevant to the reader? Please remove.

P9 L208 Did you use phonation for the speech evaluation scale? Also, the IINFVo typically uses standardized text, not prolonged vowels or seriations. If you used these for evaluation, the study is significantly flawed. Please clarify which of the recorded speech samples you used.

P9 L209 The full name is used at the first mention, and then the abbreviation is used throughout the text. Please revise and keep consistency throughout the text.

P9 L210 The experience is essential. Please mention the time (e.g., experience greater than ten years)

P9 L211This is excellent, but if you used only SV samples without healthy samples, how did your referees establish the normal voice/speech baseline? Was the prior instruction and explanation provided?

P9 L214 You reported including 89 patients in your study. What happened to the rest?

P9 L217 IINFVo

P9 L218 VAS or Likert-based? Please clarify.

P9 L208-242 It would make more sense to describe the IINFVo after the questionnaires and only then write about the evaluation. Please move the entire segment before the speech recordings section.

P9 L219 Evaluation?

P9 L220 I think you forgot to mention the intelligibility below. If you did not use this parameter, please refer to 4, and if you used it, please describe it below as suggested by Moerman

P9 L231 Quality of voicing – please refer to the original article by Moerman for clarification

P9 L234 You should not report the results in the materials and methods section. Please consider moving or removing it.

 The graph should be understandable without explanation from the text (please either expand or delete the IINFVo abbreviation)

The graph format is wrong. Your data would be better presented in a table with included SD or a column chart with provided CI. Also, why would you not report on the total IINVFo score?

P10 L239 Overexplaning. Please remove.

P10 L246 Intra-rater and inter-rater – please use consistently throughout the text

P10 L246 You mention that IINFVo is already validated in your native language by Shindler et al. [ref. 32]. I understand the need to evaluate intrarater-reliability, but why did you need to do interrater reliability again, especially when you did not report on this later? Please elaborate on this in the materials and methods section.

P10 L252 The full name is used at the first mention, and then the abbreviation is used throughout the text. Please revise and keep consistency throughout the text.

P10 L252 The sentence is unclear. Please clarify.

P10 L258 I suspect you are talking about the linear regression here. Please clarify. Furthermore, you describe this elaborated method of creating a model, but you do not present it in the results. Please either provide the product of this analysis or delete this paragraph. The same goes for ANOVA and correlation results – you describe the methods used but do not report the findings in the results section.

P10 L265 A reference would be adequate.

RESULTS This is the major flaw of your article. All your results are descriptive. And you do not report any statistically significant differences or similarities between the four groups in your results section. Also, you mentioned that you did ANOVA, correlation, and cluster analysis but did not report those findings either.

I would suggest you rewrite the results section and provide the averages, SD, and p.

Also, the correlation is in the aim of your study, so please report those or change the purpose of the study.

Figure 3 The graph should be understandable without explanation from the text (please either expand or delete the SECEL abbreviation)

The graph format is wrong. Your data would be better presented in a table with included SD or a column chart with provided CI. Also, why would you not report on the total SECEL score?

P13 L313-324 This was as VAS scale. Please report the scores with at least one number after colon with and with SD (e.g., 5.2 (SD=0.5))

P13 L322 About? Please report the exact data.

DISCUSSION Your discussion is mainly about the results (which should be in a result section). Please rewrite the discussion to overview the context of your findings in the available data (did you find anything new? How does your work compare to the other published articles? What are the shortcomings of your approach?)

P13 L332 Please report results in the result section.

P14 L345 But you did not report the correlations in the results section? Please either report them, only then make these claims

P14 L352 Reference?

P14 L360 Reference? Did you compare it with other possible tools?

TABLE 4 This is great, but did you get a statistical difference between the groups?

P15 L368 P value?

P16 L391 Great analysis, but it would be more interesting to get a better explanation. Furthermore, it seems that almost all IINFVo parameters have statistical significance, so wouldn't it be more reasonable to calculate the correlation for the total IINFVo score instead of counting them separately?

P14 L362-395 All of these are results and would be better suited in the results section.

CONCLUSIONS The first two paragraphs are better suited for the discussion.

Conclusions should rarely have references and only conclude the results of the current work or its impact regarding other publications.

P18 L410 How can you claim this if you did not perform/report IINFVo correlations with the patient's self-reported questionnaires?

Your currently reported analysis shows that different IINFVo parameters correlate with different voice rehabilitation types (which is already known and could be mentioned in the discussion)

REFERENCES Some of the references are never used in the text.

Some of the references contain hyperlinks or doi, while others do not.

Please revise and keep the reference style uniform and adhere to the journal's requirements.

Author Response

P1 L13 Abbreviation first used in the text

P1 L15 Abbreviation first used in the text

P1 L26 This is better suited for discussion, as your team searched for correlations between the patient's different psycho-perceptual parameters and vocal function perception. OK

P1 L33 But it is not the only approach. Please refer to other treatment modalities as well. I suggest using NCCN guidelines for the reference.  Maybe you’re right but, for our study purposes we prefer to select the most widely used rehabilitation technique, therefore, we described only them.

P1 L45 This is a profound statement. A reference would be advisable. ADDED IN THE DISCUSSION

P2 L47 The sentence is too long and hard to follow. Please consider splitting it into two. OK

P2 L49 More commonly abbreviated as tracheoesophageal prosthesis (TEP). I advise changing and using consistently throughout the text. OK

P2 L50 Please revise: REVIEWED

TEP prevents saliva or food from entering the lower respiratory tract and enables air to enter the esophagus, allowing the pharyngoesophageal (PE) segment to vibrate.

P2 L57 The air is voluntarily sucked in and not administered. Please revise. REVIEWED

P2 L58 Vibrations.  REVIEWED

P2 L59 As described in your reference, the electrolarynx produces vibrations but not sound. Please revise the sentence not to confuse the reader. REVIEWED

P2 L62 What does this abbreviation refer to? REVIEWED

P2 L66 Did you mean substitution voice? YES

P2 L67 Could you provide a reference?  WE ALREADY DID

P2 L73 Use the name of the scale first and the abbreviation afterward. Please keep consistency throughout the text.  OK

P2 L77 Are you writing about voice or speech? Please correct accordingly.  Impression is strongly correlated to voice parameter (intensity and tonality) and fluency

P2 L92 Please revise from 44 to 81 years. REVIEWED

P2 L92 Some of the patients might have had both, and it is known that combined treatment modalities have a more significant impact on QoL. Please elaborate   REVIEWED

P2 L94 Were more patients included in the study, and where these patients excluded or not recruited for this study? Please edit the sentence to make it clearer. Also, did you consider not including patients with neurodegenerative conditions (e.g., hearing loss and poor motor function) – disclose this information either way.  Patients with severe neurological impairment (event hearing disabilities not rehabilitated) were not enrolled in this study

P3 L99 the name first and the abbreviation afterward. Please keep consistency throughout the text. DONE

Table 1 mentions 75 males, but 77 are reported. Please revise. REVIEWED (typing error)

Table 1 You mention 14 females, but only 12 are reported. Please revise. REVIEWED (typing error)

Table 1 You do not mention smoking or XRT anywhere further in the text. Please consider if this data is relevant to the reader in the context of this article.  IRRILEVANT

 P3 L109 Was the explanation provided before administering the questionnaire? Please mention either way. DONE

P3 L112 The full name is used at the first mention, and then the abbreviation is used throughout the text. Please revise and keep consistency throughout the text. REVIEWED

P3 L112 Questionnaires are typically presented as supplementary material. Please reconsider. OK

P4 L130 Quality of life? YES

P4 L136 You mentioned that the second section evaluates the patient's vocal outcomes. Swallowing is not one of them. Please clarify. The UES and LES functions are altered after laryngectomy, and GERD symptoms are easier to occur especially in esophageal speakers. Therefore, on our opinion, quality of voice is correlated with pharygoesophageal mucosa conditions

P5 L130 Overexplaining, especially with the questionnaire provided. Please reduce or consider removing this part. DONE

P5 L140 Is it a questionnaire or a scale? Please revise and use consistently throughout the text. OK

P5 L140 Questionnaires are typically presented as supplementary material. Please reconsider. OK

P5 L147 Is it a questionnaire or a scale? Please revise and use consistently throughout the text. OK

P5 L109 This is better suited for introduction or discussion. Please consider moving. OK

P6 L150 of

P6 L154 This is better suited for introduction or discussion. Please consider removing it. OK

P8 L1161-176 The paragraphs should rarely consist of a single long sentence. Please consider shortening the explanation and providing one consistent paragraph. OK

P8 L177 Were do you recording speech or voice or both?  For better judgement on impression and fluency

P8 L178 You recorded both speech and phonation, 'The audio samples' would be more appropriate OK

P8 L178 It would be helpful to know the exact model of the microphone if other researchers try to replicate your research. DONE

P8 L186 This is not very silent. Please revise. We did as Schindler did in his work

P8 L188 The IINFVo scale evaluates speech. Why did you need to record phonation? Please explain or remove it.  We use INFVo scale. We always recorded phonation for further evaluation and for residents teaching purposes.

P8 L190 It would be helpful to know the text in your native language and translated if other researchers try to replicate your research.  (ROBERTO) Of course it was translated, but until now we do not publish the version

 ********P8 L192-207 You never mention this again in the text or use it with the IINFVo scale. How is this relevant to the reader? Please remove. We use INFVo scale

******P9 L208 Did you use phonation for the speech evaluation scale? Also, the IINFVo typically uses standardized text, not prolonged vowels or seriations. If you used these for evaluation, the study is significantly flawed. Please clarify which of the recorded speech samples you used.  Schindler’s protocol was used

P9 L209 The full name is used at the first mention, and then the abbreviation is used throughout the text. Please revise and keep consistency throughout the text. OK

P9 L210 The experience is essential. Please mention the time (e.g., experience greater than ten years) DONE

P9 L211 This is excellent, but if you used only SV samples without healthy samples, how did your referees establish the normal voice/speech baseline? Was the prior instruction and explanation provided? We relied on the great experiences of our judges, of course. There is no need to have healthy samples because the patients are not valuated by naïve listeners

 P9 L214 You reported including 89 patients in your study. What happened to the rest? We enrolled only 89 patients since the beginning, after the above mentioned selection.

 P9 L217 IINFVo. We use the INFVo scale, not IINFVo. It has been our choice

P9 L218 VAS or Likert-based? Please clarify. Please, it is already written in the text (VAS scale)

P9 L208-242 It would make more sense to describe the IINFVo after the questionnaires and only then write about the evaluation. Please move the entire segment before the speech recordings section. OK

P9 L219 Evaluation? YES

P9 L220 I think you forgot to mention the intelligibility below. If you did not use this parameter, please refer to 4, and if you used it, please describe it below as suggested by Moerman. I supposed it was implied from the beginning. We adopted the INFVo scale and not the IINFVo, because there is a strong correlation between impression and intelligibility, as stated in the original Moerman’s article 2006

P9 L231 Quality of voicing – please refer to the original article by Moerman for clarification It is already specified in the text referring to the Moerman’s article 2006

P9 L234 You should not report the results in the materials and methods section. Please consider moving or removing it.  WE THINK ABOUT IT

 The graph should be understandable without explanation from the text (please either expand or delete the IINFVo abbreviation)  We do prefer to leave the graphs legends in this way. We used INFVo scale

The graph format is wrong. Your data would be better presented in a table with included SD or a column chart with provided CI. Also, why would you not report on the total IINVFo score?  Because the INFVo scale do not imply evaluation of the total score

P10 L239 Overexplaning. Please remove. ?????

P10 L246 Intra-rater and inter-rater – please use consistently throughout the text   P10 L246 You mention that IINFVo is already validated in your native language by Shindler et al. [ref. 32]. I understand the need to evaluate intrarater-reliability, but why did you need to do interrater reliability again, especially when you did not report on this later? Please elaborate on this in the materials and methods section.  

COULD YOU PLEASE CLARIFY. We evaluated the judges abilities and not the reliability of the INFVo scale, as Schindler has done

P10 L252 The full name is used at the first mention, and then the abbreviation is used throughout the text. Please revise and keep consistency throughout the text. OK

******P10 L252 The sentence is unclear. Please clarify. PLEASE, SPECIFY THE SENTENCE

P10 L258 I suspect you are talking about the linear regression here. Please clarify.

 ON PAG.9 WAS ALREADY REPORTED THAT “The mathematical objective of DA is to weight and linearly combine the discriminating variables in some fashion so that the four groups are forced to be as statistically distinct as possible.” THIS MEANS THAT THE MODEL IS BASED ON A LINEAR COMBINATION of VARIABLES AND WEIGHTS. ON PAG.13 WAS ALREADY OUTLINED THAT THE MODEL IS A REGRESSION. ANYWAY, THE WORD “LINEAR” IS ADDED TO THE WORD REGRESSION.

***** Furthermore, you describe this elaborated method of creating a model, but you do not present it in the results. Please either provide the product of this analysis or delete this paragraph.

DONE ADDED THE MODEL

The same goes for ANOVA and correlation results – you describe the methods used but do not report the findings in the results section

All the aforesaid statistical analyses were ALREADY reported in the results paragraph of paper: e.g., ANOVA on pag.11 line 274, on pag.12 line 291, and on pag.13 line 311. The correlation matrix was reported on pag.16 Table 9.

P10 L265 A reference would be adequate. DONE

RESULTS This is the major flaw of your article. All your results are descriptive. And you do not report any statistically significant differences or similarities between the four groups in your results section. Also, you mentioned that you did ANOVA, correlation, and cluster analysis but did not report those findings either.

All the aforesaid statistical analyses were ALREADY widely reported in the results paragraph: e.g., ANOVA on pag.11 line 274, on pag.12 line 291, and on pag.13 line 311. The correlation matrix was reported on pag.16 Table 9. The discriminant analysis was reported from pagg. 13 to 15. The statistically significant differences or similarities between groups were reported on pag.15 Table 8. (see also the ANOVA test mentioned above).

I would suggest you rewrite the results section and provide the averages, SD, and p.

The averages, SD, and were ALREADY extensively reported in the paper where needed.

Also, the correlation is in the aim of your study, so please report those or change the purpose of the study.

The correlation matrix was ALREADY reported on pag.16 Table 9 (now Tab.6). 

 Figure 3 The graph should be understandable without explanation from the text (please either expand or delete the SECEL abbreviation)  DONE CHANGED THE LABEL OF Y AXIS

The graph format is wrong. Your data would be better presented in a table with included SD or a column chart with provided CI. Also, why would you not report on the total SECEL score? INFVo scale do not imply evaluation of the total score

 P13 L313-324 This was as VAS scale. Please report the scores with at least one number after colon with and with SD (e.g., 5.2 (SD=0.5))  DONE

P13 L322 About? Please report the exact data. DONE

DISCUSSION Your discussion is mainly about the results (which should be in a result section). Please rewrite the discussion to overview the context of your findings in the available data (did you find anything new? How does your work compare to the other published articles? What are the shortcomings of your approach?)

P13 L332 Please report results in the result section. DONE

P14 L345 But you did not report the correlations in the results section? Please either report them, only then make these claims

P14 L352 Reference?

P14 L360 Reference? Did you compare it with other possible tools?

TABLE 4 This is great, but did you get a statistical difference between the groups? PLEASE, INDICATE WHICH TAB ARE YOU REFERRING TO?

P15 L368 P value? WHERE?

P16 L391 Great analysis, but it would be more interesting to get a better explanation. Furthermore, it seems that almost all IINFVo parameters have statistical significance, so wouldn't it be more reasonable to calculate the correlation for the total IINFVo score instead of counting them separately? IT IS NOT TRUE, because from our analysis the parameter F is statistically irrelevant

P14 L362-395 All of these are results and would be better suited in the results section. DONE

CONCLUSIONS The first two paragraphs are better suited for the discussion.

Conclusions should rarely have references and only conclude the results of the current work or its impact regarding other publications.

 *****P18 L410 How can you claim this if you did not perform/report IINFVo correlations with the patient's self-reported questionnaires? The correlation matrix was ALREADY reported as Table 9 (now Tab.6).

Your currently reported analysis shows that different IINFVo parameters correlate with different voice rehabilitation types (which is already known and could be mentioned in the discussion) DISCUSSED

REFERENCES Some of the references are never used in the text. CORRECTED

Some of the references contain hyperlinks or doi, while others do not. CORRECTED

Please revise and keep the reference style uniform and adhere to the journal's requirements. DONE

Reviewer 3 Report

The manuscript reported subjective and perceptual evaluation on alaryngeal voice with a focus on the impact of quality of life.  A few selected questions from Voice-Related Quality of Life (V-RQoL) and the SECEL scale were selected.  Acoustic recording was collected for perceptual evaluation using INFVo.   A good number of patients (89) were recruitment and divided into four groups, TEP, EL, ES, and voiceless. Comparing these four groups is interesting.  I don’t have any major concerns in the general experimental design.  There were, however, a major concern and other additional minor to moderate comments.

My major concern is on the writing. This manuscript is hard to follow due to some important information is missing (also partly due to so many tracked changes)

First, Table 2 should be in supplement material, as the analysis just focused on only one question.       (A related minor point) Table 2 It looks like a picture (the text is fuzzy).  It is better to copy and paste the content (or type) in the supplement materials.

Line 124-134, this paragraph on description of V-RQoL can be condensed, as only the first question was used in the analysis.  

Table 3 should be in a supplement document as well.   Some specific question can be embedded as examples in the description of these subscales (line 163 to 177).

what specific scales/measures were used in analysis can be explicitly mentioned in the end of section 2.3. 

Second, what exactly were the input variables for the discriminant analysis?

Below are just minor to moderate comments.

Line 72, it seems the acronym INFVo represents Impression, Intelligibility, nose, fluency, and Voice.  It can be simply spelled out in parathesis after INFVo ,rather than putting them after “i.e.”.

Line 197,  it may be good to change the title of section 2.5 to include more important information, for example,  Speech samples for perceptual evaluation.   There was no acoustic analysis in this study. Thus, it’s good to highlight that these speech recordings were used listening judgement.

Line 198, what are the brand and model of the microphone?

Line 231, were both evaluations used in the analysis or just one of them (and which one)?

What’s the figure about at line 253?    Caption and title for Y axis are missing. Or is it marked as deleted (can’t tell it in the pdf)?

Figure 1, y axis title of the top panel (or is it marked deleted?) is missing.  A sub-caption is also needed.

Figure 1, bottom picture, what are the numbers (e.g., 24,836,   27,615,   21,25, and 31, 647)?

Line 304, not sure why the question description is marked as deleted.  It’s good to have the question description here.  Such slight redundancy can facilitate reading.  

Figure 2, 100% of the EL users think their speaking voice is excellent?  This is surprising to me, as it is known ELS tend to be have robotic-like voice.

Figure 4, it’s good to have the item 35 in the caption to facilitate the reading.

Captions of tables are sometimes before the table and sometimes after the table (e.g., line 383 and line 387), which needs to be consistent.

Line 456, a word change may be needed.  I don’t think INFVo is “proven” to be useful. It may be “demonstrated” to be a useful tool, for example.

Author Response

REVISORE 3

The manuscript reported subjective and perceptual evaluation on alaryngeal voice with a focus on the impact of quality of life.  A few selected questions from Voice-Related Quality of Life (V-RQoL) and the SECEL scale were selected.  Acoustic recording was collected for perceptual evaluation using INFVo.   A good number of patients (89) were recruitment and divided into four groups, TEP, EL, ES, and voiceless. Comparing these four groups is interesting.  I don’t have any major concerns in the general experimental design.  There were, however, a major concern and other additional minor to moderate comments.

My major concern is on the writing. This manuscript is hard to follow due to some important information is missing (also partly due to so many tracked changes)

First, Table 2 should be in supplement material, as the analysis just focused on only one question.       (A related minor point) Table 2 It looks like a picture (the text is fuzzy).  It is better to copy and paste the content (or type) in the supplement materials.

Line 124-134, this paragraph on description of V-RQoL can be condensed, as only the first question was used in the analysis.  

Table 3 should be in a supplement document as well.   Some specific question can be embedded as examples in the description of these subscales (line 163 to 177).

what specific scales/measures were used in analysis can be explicitly mentioned in the end of section 2.3. 

 Second, what exactly were the input variables for the discriminant analysis? ALL THE VARIABLES, NAMELY: I, N, F, Vo, SECEL_TOT, VRQoL_TOT, QUESTION A, ITEM 35, AGE.

   Below are just minor to moderate comments.

 Line 72, it seems the acronym INFVo represents Impression, Intelligibility, nose, fluency, and Voice.  It can be simply spelled out in parathesis after INFVo ,rather than putting them after “i.e.”. OK

Line 197,  it may be good to change the title of section 2.5 to include more important information, for example,  Speech samples for perceptual evaluation.   There was no acoustic analysis in this study. Thus, it’s good to highlight that these speech recordings were used listening judgement.

 Line 198, what are the brand and model of the microphone? OK

Line 231, were both evaluations used in the analysis or just one of them (and which one)?

What’s the figure about at line 253?    Caption and title for Y axis are missing. Or is it marked as deleted (can’t tell it in the pdf)?

 Figure 1, y axis title of the top panel (or is it marked deleted?) is missing.  A sub-caption is also needed.

 Figure 1, bottom picture, what are the numbers (e.g., 24,836,   27,615,   21,25, and 31,647)?

 ADDED CAPTION

 Line 304, not sure why the question description is marked as deleted.  It’s good to have the question description here.  Such slight redundancy can facilitate reading.  

Figure 2, 100% of the EL users think their speaking voice is excellent?  This is surprising to me, as it is known ELS tend to be have robotic-like voice.

 They judge positively (sorry, typing error, not excellent), even though these patients are a limited cohort and this is surely a limitation for the study. DISCUSSION WIDENED.  By the way, referring to my almost forty-year-experience on laryngectomized patient I think that is a world apart and many other factors could contribute to a positive quality of life. I have a woman with a wonderful TE voice and she always asked for a larynx transplant, wheraeas many ELS have no problems with their voice. GOD ONLY KNOWS!

Figure 4, it’s good to have the item 35 in the caption to facilitate the reading. OK

Captions of tables are sometimes before the table and sometimes after the table (e.g., line 383 and line 387), which needs to be consistent.

 Line 456, a word change may be needed.  I don’t think INFVo is “proven” to be useful. It may be “demonstrated” to be a useful tool, for example. AKNOWLEDGED
